# A Narrative Review of the IL-18 and IL-37 Implications in the Pathogenesis of Atopic Dermatitis and Psoriasis: Prospective Treatment Targets

**DOI:** 10.3390/ijms25158437

**Published:** 2024-08-02

**Authors:** Lluís Rusiñol, Lluís Puig

**Affiliations:** 1Dermatology Department, Hospital de la Santa Creu i Sant Pau, 08025 Barcelona, Spain; lrusinolb@gmail.com; 2Institut de Recerca Sant Pau (IR Sant Pau), Sant Quintí 77-79, 08041 Barcelona, Spain; 3Unitat Docent Hospital Universitari Sant Pau, Universitat Autònoma de Barcelona, 08193 Bellaterra, Spain

**Keywords:** Psoriasis, atopic dermatitis, IL-18, IL-37

## Abstract

Atopic dermatitis and psoriasis are prevalent inflammatory skin conditions that significantly impact the quality of life of patients, with diverse treatment options available. Despite advances in understanding their underlying mechanisms, recent research highlights the significance of interleukins IL-18 and IL-37, in Th1, Th2, and Th17 inflammatory responses, closely associated with the pathogenesis of psoriasis and atopic dermatitis. Hence, IL-18 and IL-37 could potentially become therapeutic targets. This narrative review synthesizes knowledge on these interleukins, their roles in atopic dermatitis and psoriasis, and emerging treatment strategies. Findings of a literature search up to 30 May 2024, underscore a research gap in IL-37-targeted therapies. Conversely, IL-18-focused treatments have demonstrated promise in adult-onset Still’s Disease, warranting further exploration for their potential efficacy in psoriasis and atopic dermatitis.

## 1. Introduction

Interleukin (IL)-18 and IL-37 are constituents of the IL-1 cytokine family, which is divided into three subgroups: IL-1 (including IL-1α, IL-1β, IL-33, and IL-1Rα), IL-18 (comprising IL-18 and IL-37), and IL-36 (composed of IL-36α, IL-36β, IL-36γ, IL-36Rα, and IL-38) [1,2,3]. IL-18, known for its inflammatory properties, is notably abundant in epithelial barrier cells of the skin, upper airway, and gastrointestinal tract [4,5,6,7]. Recent research has underscored its emerging role in pathogenesis and clinical contexts [8]. Conversely, IL-37, a more recently identified cytokine, possesses anti-inflammatory properties [9]. Its synthesis is stimulated by pro-inflammatory agents, acting as a safeguard against excessive inflammation and subsequent tissue damage, thereby modulating both innate and adaptive immune responses [10]. It is noteworthy that, unlike other IL-1 family members, there is no homologous gene for IL-37 in mice, necessitating the use of transgenic mouse models for its study [9]. Both IL-18 and IL-37 have shown abnormal expression levels in various inflammatory and autoimmune disorders, including skin diseases such as atopic dermatitis (AD) and psoriasis. Psoriasis and atopic dermatitis (AD) are chronic immune-mediated inflammatory skin conditions affecting a significant proportion of the world’s population. They carry a significant disease burden, diminishing the health-related quality of life and particularly impacting mental well-being [11,12]. Despite significant advancements in the targeted treatment of psoriasis and AD through the inhibition of key cytokines with pathogenetic relevance, considerable gaps persist in identifying optimal treatment strategies tailored to individual patients’ needs. Growing insights into the roles of IL-18 and IL-37 in these diseases may contribute to the development of newer therapeutic alternatives.

## 2. Material and Methods

An electronic literature search was conducted on the Medline/PubMed database up to May 2024, using Medical Subject Headings (MeSH) terms and relevant medical terminology. The search criteria included the terms: ‘IL-18’, ‘IL-37’, ‘psoriasis’, and ‘atopic dermatitis’. We considered original studies, reviews, systematic reviews, and meta-analyses that specifically investigated the role of IL-18 and IL-37 in AD and psoriasis. Manuscripts written in English were eligible for inclusion, while letters to the editor, editorials, expert opinions, conference proceedings, and studies exclusively involving patients with psoriatic arthritis were excluded. The selection of publications was performed by two independent researchers (LR, LP), and any disagreements were resolved through consensus.

## 3. Atopic Dermatitis and Psoriasis Pathogenesis

AD is one of the most prevalent chronic cutaneous immune-mediated inflammatory disease, affecting approximately 10% of adults and 20% of children globally [11]. Its pathogenesis is multifaceted, involving genetic susceptibility, epidermal dysfunction, and T-cell-mediated inflammation [13,14,15,16,17]. Both innate and adaptive immune responses play significant roles in AD etiology [13,15]. Traditionally viewed as primarily driven by T-helper 2 (Th2) responses, recent research reveals a more complex immunological landscape, involving activation of Th2, Th22, Th17, and Th1 subsets [13,14,15]. Notably, a dynamic transition from Th2 to Th1 responses occurs in both acute and chronic AD lesions. Key cytokines such as interleukin-4 (IL-4) and interleukin-13 (IL-13) are central to AD pathophysiology, promoting Th2 cell differentiation and IgE production [18,19]. Elevated levels of Th2-related cytokines exacerbate inflammation in acute AD lesions and contribute to epidermal barrier dysfunction by downregulating terminal differentiation proteins [18]. Understanding the intricate interplay between cytokine dysregulation, immune activation, and barrier dysfunction is crucial for elucidating AD mechanisms and guiding therapeutic strategies to improve patient outcomes.

Psoriasis, a chronic immune-mediated inflammatory skin disorder, has a prevalence ranging from 1% to 3% in Western populations. The pathogenesis of psoriasis is characterized by a complex interplay between keratinocytes and the innate and adaptive immune systems [12,20,21]. Various hypotheses have been proposed to elucidate the underlying mechanisms driving psoriasis pathogenesis [20,22,23]. The prevailing theory posits an initial triggering event followed by a sustained chronic inflammatory phase orchestrated by a positive feedback loop mediated by cytokine-induced keratinocyte activation and proliferation [24,25]. In genetically predisposed individuals, exposure to triggers such as trauma, infection, or certain medications precipitates keratinocyte apoptosis or necrosis, leading to the release of nucleic acids (DNA or RNA). These nucleic acids can activate a type I interferon-mediated autoinflammatory response, along with potential autoantigens including cathelicidin (LL37), ADAMTSL-5, and phospholipase A2 (PLA2G4D). LL37 interacts with self-DNA or -RNA to trigger innate immune responses via Toll-like receptors (TLR), notably TLR7 and TLR8 [26,27]. Activation of these receptors induces the production of interferon (IFN)-α and IFN-β by keratinocytes and plasmacytoid dendritic cells (pDCs), as well as interleukin (IL)-6 and tumor necrosis factor (TNF) by myeloid dendritic cells (mDCs). IL-6 promotes the differentiation of CD4+ naïve T-cells into T helper (Th)-17 cells [28,29], while type I IFNs and TNF facilitate the secretion of IL-12 and IL-23 by mDCs [30,31]. IL-12 and IL-23 play pivotal roles in orchestrating the immune cascade underlying psoriasis pathogenesis. IL-12 drives the differentiation of CD4+ naïve T-cells into Th1 cells in conjunction with tumor growth factor (TGF)-β and IL-6, while IL-23 promotes Th17 cell development and activates αβ T cells [32,33,34,35,36]. Th1 and Th17 activated cells exhibit distinct cytokine secretion profiles, with Th1 cells producing IFN-γ and TNF, and Th17 cells releasing IL-17, IL-22, and TNF [37]. These cytokines collectively induce keratinocyte proliferation, differentiation, and inflammatory activation, with IL-17A serving as the principal effector cytokine driving psoriasis pathogenesis and being essential for the development and maintenance of psoriatic plaques [37]. Additionally, stimulated keratinocytes produce inflammatory cytokines such as IL-1, IL-6, and TNF, as well as chemokines including CXCL1, CXCL2, and CXCL3, and antimicrobial peptides (AMPs) such as S100A7/8/9, human β-defensin 2, and LL-37 [35,38,39]. These mediators attract and activate immune cells, exacerbating psoriatic inflammation.

## 4. Biology of IL-18

As constituents of the IL-1 family, IL-18 and IL-37 are synthesized and initially present in the cytoplasm in an inactive precursor form, awaiting subsequent activation through structural modifications [8]. The IL-18 gene is located on chromosome 11 [40].

### 4.1. Synthesis of IL-18

Various inflammatory stimuli, such as lipopolysaccharide (LPS), interferons (IFNs), and IL-22, prompt transcription and translation of the IL-18 gene [41,42,43]. This results in the synthesis of pro-IL-18, which is subsequently released into the cytoplasm, where it undergoes activation. The process of IL-18 activation is intricate and remains incompletely understood. Pro-IL-18, which lacks a well-defined biological function, undergoes cleavage mediated by inflammasome-mediated activation of caspase-1, a process also involved in the activation of IL-1β [44,45,46]. Notably, several inflammasomes, including NLRP3, pyrin, NLRC4, NLRP6, and NLRP9B, play roles in IL-18 activation [47,48,49]. Genetic associations have been established between systemic IL-18 levels and NLRC4 activity [48]. Additionally, non-inflammasome proteases can activate IL-18; for instance, Fas ligand stimulates macrophages, leading to the release of active IL-18, while granzyme-B from cytotoxic granules cleaves keratinocyte-derived pro-IL-18 [50,51]. Furthermore, caspase-1 can cleave gasdermin-D, creating additional pores through which IL-18 can exit [52,53,54]. Among its diverse functions, IL-18 mediates IFN-γ production [55]. Eventually, IL-18 is transported to the extracellular space through cellular pores [56]. IL-18 is produced by macrophages, and nearly all barrier epithelia harbor a significant reservoir of pro-IL-18.

Extracellular IL-18 exhibits a half-life of approximately 16 hours [57]. Its activity is counteracted by the presence of the IL-18 binding protein (IL-18BP), which has a high affinity for IL-18 and solely functions to inhibit IL-18 activity [58]. In healthy individuals, over 95% of circulating IL-18 is bound to soluble IL-18BP [58,59]. Given the strength of this interaction, it is unlikely that IL-18BP serves as a reservoir for cleaved IL-18 [8,60]. Notably, IFN-γ serves as a potent inducer of IL-18BP, leading to an indirect negative-feedback loop involving IL-18, IFN-γ, and IL-18BP [61,62]. Furthermore, immune evasion and active suppression by neoplasms and various viruses, including poxviruses, Chikungunya, and hepatitis C, involve the induction of IL-18BP expression [58,63,64,65]. IL-37 also influences extracellular IL-18 function, dampening its inflammatory effects, by binding to IL-18R [8,60].

### 4.2. Signaling of IL-18

IL-18 signaling is facilitated through the IL-18 receptor (IL-18R), composed of two chains referred to as α and β, encoded by the *IL18R1* and *IL18RAP* genes, respectively [66]. Initially, IL-18 binds to the α chain with modest affinity, followed by the recruitment of the β chain, leading to the assembly of a high-affinity receptor capable of initiating signaling cascades [67,68,69,70,71] (Figure 1). The cytosolic region of the IL-18R includes a Toll/IL-1 receptor (TIR) domain, where myeloid differentiation primary response 88 (Myd88) binds, leading to the recruitment and phosphorylation of IL-1 receptor-associated kinases (IRAKs) [72,73,74]. Notably, IRAK1 and IRAK4 have been implicated in IL-18 signaling in Th1 and NK cells. These IRAKs further associate with the adaptor molecule TNF receptor-associated factor 6 (TRAF6), which in turn binds to the kinase TAK1 (transforming growth factor-b-activated kinase 1) [56]. TAK1 phosphorylates nuclear factor kappa B (NF-κB)-induced kinase (NIK), leading to NIK activation. Subsequently, NIK activates the inhibitor of nuclear kappa B (IkB) kinase (IKK), which phosphorylates IkB, resulting in IkB ubiquitination and rapid degradation. The freed transcription factor NF-κB translocates into the nucleus, where it binds to specific regulatory sequences (κB sites) in the promoter regions of various inflammatory genes (e.g., IFN-γ, IL-8, IL-1β, TNF-α) [75,76,77,78,79]. Moreover, TRAF6 phosphorylation also associates with apoptosis signal-regulating kinase 1 (ASK1) through TAK1, activating mitogen-activated protein kinases (MAPK)-related signaling downstream, including p38 mitogen-activated protein kinase (p38MAPK), Jun kinase (JNK), phosphoinositide 3-kinase (PI3K)/AKT, and extracellular regulated protein kinases (ERK), ultimately leading to the activation of the AP-1 transcription factor [78,80] (Figure 2).

The expression of genes encoding the α and β chains of IL-18R is predominantly confined to immune cells, particularly natural killer (NK) cells, NK-like innate lymphoid cells (ILCs), and activated CD4+ and CD8+ T cells [8]. While other immune cell types such as B cells, γδ-T cells, basophils, dendritic cells (DCs), synovial macrophages, mast cells, neutrophils, and skin-resident type 2 innate lymphoid (ILC2) cells also express *IL18R1* and *IL18RAP*, their functional significance remains unclear [8,81,82,83,84,85,86,87].

According to Hu et al., naïve T cells exhibit minimal expression of IL-18R genes initially. However, following antigen presentation or stimulation with inflammatory cytokines such as IL-2, IL-12, and IL-15, the expression of *IL18R1* and *IL18RAP* is induced [88]. Additionally, T cell receptor (TCR) signaling is essential for IL-18 to interact with CD8+ T cells, indicating that the effects of IL-18 on these cells are context-dependent [89]. Conversely, NK cells, NK T cells, and NK-like ILCs constitutively express both *IL18R1* and *IL18RAP* genes, highlighting the significant role of those cells in contributing to the alarmin function of IL-18.

IL-18 signaling has numerous functions, primarily supporting the initiation and maintenance of cellular states associated with IFN-γ production [8]. IL-18BP inhibits the biological activity of IL-18, thereby reducing IFN-γ production and dampening Th1 immune responses. An imbalance in the IL-18/IL-18BP ratio may contribute to the perpetuation of chronic inflammation driven by T helper type I cytokines [56,90,91].

### 4.3. IL-18 in Keratinocytes

Keratinocytes constitute approximately 95% of the epidermal layer and, beyond their structural role, significantly contribute to skin inflammation and immune responses [56]. Compared to monocytes, peripheral blood mononuclear cells (PBMCs), or leukocytes, keratinocytes synthesize substantial amounts of pro-IL-18 [6]. Cho et al. demonstrated the induction of IL-18 production in the human keratinocyte cell line HaCaT following ultraviolet B (UVB) irradiation [92]. This induction occurred in a time- and dose-dependent manner, facilitated by the generation of reactive oxygen intermediates (ROIs) and the activation of activator protein-1 (AP-1). Notably, keratinocytes express the IL-18 receptor (IL-18R), suggesting that IL-18 may exert its effects in an autocrine or paracrine fashion on neighboring keratinocytes [93]. According to Fenini et al., NRLP1 inflammasomes play a crucial role in sensing UVB radiation and subsequently facilitating the secretion of IL-1β and IL-18 from keratinocytes [94].

IFN-γ induces the expression of chemokines such as CXCL9 (chemokine C-X-C motif ligand 9), CXCL10, and CXCL11. These chemokines recruit Th1 cells by binding to CXCR3 on the cell surface, resulting in significant infiltration of Th1 cells observed in inflammatory skin conditions [95]. In keratinocytes, IL-18 amplifies the mRNA expression of CXCL9, CXCL10, and CXCL11 induced by IFN-γ [96]. Additionally, IFN-γ activates signal transducer and activator of transcription 1 (STAT1) via Janus kinase 1 (JAK1)/JAK2 and/or p38MAPK pathways, leading to the production of CXCL9, CXCL10, and CXCL11. IFN-γ may also influence the activation of interferon regulatory factor-1 (IRF-1), responsible for CXCL11 expression, primarily through the p38MAPK pathway. IL-18 enhances this activation via PI3K/AKT and MEK/ERK pathways. IL-18 induces NF-kB activity through MEK/ERK and PI3K/AKT pathways, thereby augmenting IFN-γ -induced CXCL9 secretion [96].

Furthermore, IL-18 has been shown to possess antifibrotic effects on human dermal fibroblasts [97]. IL-18 suppresses the activation of the transcription factor Ets-1 through the ERK pathway, resulting in decreased collagen expression and inhibition of collagen production induced by transforming growth factor-beta (TGF-β).

### 4.4. IL-18 in Immune Cells

The skin, serving as the primary barrier of the body, is continually exposed to various external stimuli, leading to irritation of keratinocytes and subsequent release of IL-18. This cytokine not only interacts with neighboring keratinocytes but also reaches immune cells present in the skin, including DCs, Langerhans cells (LCs), and macrophages [56]. Upon stimulation by diverse triggers, LCs undergo activation into mature DCs. Specifically, IL-18 induces the accumulation of DCs and the migration of LCs to local lymph nodes.

Within the skin-draining lymph nodes, LCs present antigens to CD4+ T cells, thereby mediating the adaptive immune response. Although IL-18 alone does not induce Th1 cell differentiation, it promotes the expression of the interleukin-12 receptor β (IL-12Rβ), thereby facilitating IL-12-mediated Th1 cell differentiation [98,99]. Additionally, the combined action of IL-18 and IL-12 synergistically stimulates Th1 cells to produce IFN-γ.

IL-18 also interacts with neutrophils and PBMCs, eliciting the release of various inflammatory cytokines in neutrophils and stimulating the production of IL-8 in PBMCs [100]. Conversely, IL-18BP inhibits the IL-12-induced production of IFN-γ in PBMCs.

IL-18 has contrasting effects on the balance between Th1 and Th2 activation. When in conjunction with IL-12 or IL-15, IL-18 promotes the differentiation of naive T cells into Th1 cells. However, IL-18 increases IgE levels and the production of IL-4 and IL-13 by mast cells, basophils, NK cells, and CD4+ T cells, thereby promoting a Th2 response [100,101,102]. IL-18, along with IL-2, enhances IL-13 production in NK cells and T cells [96]. Furthermore, IL-18 synergizes with IL-13 to trigger histamine release from basophils. IL-18 also enhances the expression of Fas Ligand on NK cells, promoting Fas Ligand-mediated NK cytotoxicity [103].

IL-18 can stimulate allergic inflammation by increasing the production of key factors contributing to atopic inflammation through mast cells and basophils [104]. This evidence underscores the pleiotropic nature of IL-18, playing a pivotal role in fostering immune responses and inflammation, thereby serving as a crucial link between innate and adaptive immunity.

Moreover, IL-18 may contribute to the induction of Th17 cell responses by enhancing IL-17 production in already polarized Th17 cells in conjunction with IL-23, independently of TCR activation [105].

Kinoshita et al. observed that IL-18 injections activated B-1 cells in the liver, leading to increased production of IgM and enhanced immunity against bacterial infections following a burn injury [106].

Finally, the role of IL-18 in maintaining homeostasis is underscored by findings from studies involving IL-18 deficient mice, which demonstrated a predisposition to obesity and other metabolic disorders. These mice exhibited a substantial increase in body weight (by 40%) and body fat content (over 100%) compared to wildtype animals. Similarly, individuals with impaired expression of the IL-18Rα receptor on their cell surfaces also displayed susceptibility to obesity, diabetes, and other metabolic disorders [105,107].

## 5. Biology of IL-37

IL-37, a member of the IL-1 family, was first identified in 2000 and its gene is located on chromosome 2q12-13 [108]. Unlike other cytokines within the IL-1 family, the murine homologue gene for IL-37 has yet to be discovered [9], necessitating the use of transgenic mice models for functional studies.

### 5.1. Synthesis of IL-37

IL-37 expression has been documented in multiple human tissues and cell lines. IL-37 is mainly produced by circulating monocytes, macrophages, dendritic cells, tonsillar B cells, and plasma cells, as well as by epithelial cells in the skin and gut as a response to inflammation.

Functionally, IL-37 serves as an anti-inflammatory cytokine that responds to pro-inflammatory stimuli [10]. It plays a crucial role in mitigating excessive inflammation and preventing severe tissue damage, thereby regulating both the innate and adaptive immune responses. Elevated levels of IL-37 have been consistently observed in various inflammatory and autoimmune conditions [10], and significant attention has been directed towards its potential inhibitory effects on cancer initiation and progression [109]. The IL-37 gene proximity to the regulatory regions of the IL-1α and IL-1β genes suggests a potential relationship with the anti-inflammatory role of IL-37 [108].

Despite structural and production process similarities between IL-37 and other cytokines of the IL-1 family, particularly IL-18, notable differences exist. IL-37 is secreted as a precursor form that is biologically active both in vitro and in vivo. This active precursor form of IL-37 is mainly secreted into the extracellular space. Conversely, the intracellular precursor form of IL-37 is activated by caspase-1-mediated cleavage, allowing its translocation into the nucleus [9,110]. Unlike IL-18 and other members of the IL-1 family, IL-37 is not constitutively expressed in healthy individuals’ cells; its expression is upregulated in response to pro-inflammatory stimuli [108]. Consequently, IL-37 is hypothesized to orchestrate a negative-feedback mechanism to regulate and mitigate excessive inflammation, maintaining immunological homeostasis.

IL-37 can act both intracellularly and extracellularly, though the specific conditions dictating its predominant mechanism of action remain unclear [111]. Precursor and mature forms of IL-37 are released to the extracellular space. Precursor forms are activated by proteases [9]. No specific IL-37 receptor has been identified [9], but IL-37 binds to the α chain of the IL-18 receptor (IL-18R), with about 50 times lower affinity than IL-18 [67] leading to the recruitment of the decoy receptor IL-1R8, encoded by *SIGIRR*. This extracellular mechanism blocks the IL-18 inflammatory pathway [112]. Additionally, IL-37 can bind to IL-18BP, forming a complex that binds to the β chain of IL-18R [9], preventing the formation of the functional (heterodimeric) IL-18 receptor (Figure 3).

### 5.2. IL-37 Signaling

Regarding intracellular anti-inflammatory effects, the absence of a nuclear localization sequence suggests that the transportation of IL-37 to the nucleus likely occurs through association with Smad3 [111]. IL-37 enhances the anti-inflammatory functions of Smad3 rather than directly engaging in DNA binding. Upon exposure to pro-inflammatory stimuli, the intracellular concentration of IL-37 precursor increases. Subsequently, activated caspase-1 cleaves the precursor, facilitating the binding of the C-terminal domain of IL-37 to Smad3 [111]. Following phosphorylation, this complex translocates to the nucleus, where it contributes to the regulation of gene expression [111], suppressing inflammatory pathways such as mammalian target of rapamycin (mTOR), mitogen-activated protein kinase (MAPK), NF-kB, and various transcription factors [113,114]. IL-37 also enhances the activation of STAT3, protein phosphatase and tensin homolog (PTEN), and 5’AMP-activated protein kinase (AMPK) [113].

Interestingly, at elevated concentrations, the inhibitory effect of IL-37 on inflammatory cytokines diminishes due to the formation of IL-37 homodimers, initiating an auto-regulatory mechanism that curtails its immunosuppressive effect [111].

Several stimuli have been identified as triggers for IL-37 expression or inhibition [115]. Toll-like receptor (TLR) ligands such as lipopolysaccharides (LPS) and Pam3CysSerLys4 (Pam3CSK4), as well as TGF-β1, induce IL-37 production. Conversely, Granulocyte-Macrophage Colony-Stimulating Factor (GM-CSF) and IL-4 inhibit its expression. IL-37 appears to be stored in monocytes for rapid release following stimulation by LPS, for example [116].

### 5.3. IL-37 in the Skin

While there remains a considerable need for further research on the role of IL-37 in skin biology, recent studies have provided new insights. IL-37 is predominantly expressed in mature and differentiated keratinocytes [117,118]. Multiple studies have reported decreased expression of IL-37 in the epidermis of patients with AD [117,119,120], although findings have been inconsistent [121]. Specifically, chronic AD skin lesions have shown the most significant reduction in IL-37 levels [117].

Zhou et al. established a positive association between the expression of IL-37 and that of filaggrin (FLG), FLG2, and involucrin (IVL) [117]. Additionally, using an in vitro 3D human skin model (EpidermFT), they demonstrated that IL-4, IL-13, and IL-31 can decrease IL-37 levels. Conversely, IL-37 increased the expression of FLG and FLG2 but had no impact on IVL. In a separate study, Hou et al. observed a reduction in epidermal thickness and scratching frequency in IL-37-transgenic mice [119].

IL-33 plays a well-established role in the pathogenesis of AD by inducing a type 2 immune response, which produces IL-31 that directly stimulates nerves, leading to pruritus [122]. According to Cevikbas et al. [123] IL-31 production in skin cells was found to be primarily attributed to Th2 cells, with a lesser contribution from mature dendritic cells among the immune and resident skin cell populations studied. Injection of IL-31 into the skin and spinal cord induced severe itching, and its levels notably increased in murine models of atopy-like dermatitis. Both human and mouse dorsal root ganglia neurons expressed IL-31RA, particularly in neurons co-expressing transient receptor potential cation channel vanilloid subtype 1 (TRPV1). Itch induced by IL-31 was significantly diminished in mice deficient in TRPV1 and transient receptor potential cation channel ankyrin subtype 1 (TRPA1), but not in c-kit or proteinase-activated receptor 2 mice. In vitro studies with cultured primary sensory neurons revealed that IL-31 triggered calcium release and phosphorylation of extracellular signal-regulated kinase 1/2 (ERK1/2), inhibition of which disrupted IL-31 signaling and reduced IL-31-induced scratching in vivo [123].

IL-37 was found to inhibit the expression of keratinocyte-derived IL-33, possibly through the inhibition of MAPK and the prevention of STAT1 activation in keratinocytes [122,124]. Therefore, it is reasonable to hypothesize that IL-37 may modulate epidermal barrier function [117]. However, a significant proportion of patients with AD harbor an inherent mutation in the FLG gene, suggesting that the administration of IL-37 alone may be insufficient for these patients [125].

### 5.4. IL-37 in Immune Cells

IL-37 levels are decreased in conditions such as AD, asthma, and allergic rhinitis [119,126,127,128]. In allergic patients, reduced IL-37 levels may lead to ineffective immune suppression and dysregulated responses to allergen stimulation [119].

The Th2 immune response appears to inhibit IL-37 production [129], while IL-37 itself can influence Th2 cells [130]. STAT6 activation, facilitated by IL-4, promotes the differentiation of CD4+ T cells into Th2 cells by upregulating GATA3. In animals treated with IL-37, both STAT6 phosphorylation and IL-4 levels were decreased, indicating that IL-37 potentially inhibits GATA3 expression by disrupting the IL-4/STAT6 signaling pathway. This inhibitory effect consequently reduces the proliferation, differentiation, and activation of Th2 cells, as observed in the IL-37-transgenic mouse model of allergic rhinitis [130]. Additionally, in vivo studies have shown that IL-37 can inhibit the function of not only Th2 cells but also Th1 and Th17 cells by peripheral blood mononuclear cells (PBMCs), M1 macrophages, and dendritic cells (DCs) [131]. Human DCs when cultured with CD4+ T cells and treated with IL-37 obtain a tolerogenic phenotype and promote Tregs expansion and suppression of Th1 and Th17 populations [132].

Th2-secreted cytokines (IL-4 and GM-CSF) suppress IL-37 in CD4+ T cells, while Th1-secreted cytokines (IFN-γ and IL-17) induce IL-37 expression in these cells [133].

IL-37 also mediates Treg cell activity, contributing to their production of TGF-β1 and/or IL-10 [10,134,135,136]. IL-37 expression level is consistently low in freshly isolated human CD4+CD25+ Tregs that are not stimulated [10]. Silencing IL-37 in these Tregs significantly reduces their suppressive capability [135]. Stimulation with IL-37 enhances the suppressive function of CD4+CD25+ Tregs by upregulating CTLA-4 and FOXP3 and increasing TGF- β1 levels, although it does not IL-10 levels [134].

IL-2 enhances the proliferative responses of effector T cells (Teff) and NK cells while maintaining immune homeostasis by promoting the proliferation, differentiation, and function of regulatory Tregs. Wang et al. studied the effect of IL-37 and IL-2 in mice, comparing groups pre-treated with recombinant IL-37 to non-pre-treated groups. No disparity in IL-2 levels has been found between IL-37-stimulated and control groups [134], although reduced IL-37 levels contribute to increased IL-2 secretion and exacerbated inflammation [135].

IL-37 also interacts with macrophages, inhibiting Notch1 and NF-kB activation and thereby attenuating M1 (proinflammatory) polarization; furthermore, recombinant IL-37 enhance M2 macrophages and their secretion of anti-inflammatory cytokines [137].

The expression of IL-37 in dendritic cells prevents their maturation via the IL-1R8–TLR4–NF-kB pathway and function, resulting in semi-mature tolerogenic dendritic cells that inhibit Teff activation and promote Treg development [138,139].

In the presence of IL-4 and IL-13, B cells undergo immunoglobulin class-switch recombination, facilitated by IL-4Rα and STAT6 [140]. This event prompts a transition from producing IgM to IgE antibodies. In an experimental murine model of allergic rhinitis, IL-37 administration led to decreased concentrations of IL-4 and IL-13 in serum and nasal lavage fluid, resulting in lower IgE levels [141]. However, several studies have found no statistically significant correlation between IL-37 and IgE levels in AD [119,129].

## 6. IL-18 and IL-37 in Inflammatory Skin Diseases

Despite there being much ground to cover, accumulating evidence highlights the involvement of IL-18 and IL-37 in the pathogenesis of inflammatory diseases such as AD and psoriasis.

Elevated levels of IL-18 have been detected in both the serum and lesional skin of individuals with psoriasis compared to healthy controls [142]. These heightened levels correlate with the Psoriasis Area and Severity Index (PASI) values, suggesting IL-18 as a potential biomarker for psoriasis [143,144,145]. A 2021 study by Verma et al. demonstrated increased plasma levels of IL-18 and IL-1β in psoriasis patients, which normalized with anti-TNF therapies [146]. In a mouse model of imiquimod-induced psoriasis, IL-18 exacerbated inflammation and contributed to micro-abscess formation of scale development by upregulating pro-inflammatory cytokines and reducing anti-inflammatory cytokines [147]. Nonetheless, IL-18 levels have also been suggested as potential biomarkers for multiple diseases, including: rheumatoid arthritis [84], systemic lupus erythematosus [148], inflammatory bowel disease [149], insulin resistance and metabolic syndrome [150], atherosclerosis, myocardial infarction and other cardiovascular conditions [151], as well as various cancers such as colorectal cancer, and non-small cell lung cancer [152]. Therefore, although IL-18 is not specific to psoriasis nor the sole biomarker for the condition, its increased levels in serum of patients with psoriasis suggests that IL-18 may be useful in diagnosing, monitoring, and understanding the pathogenesis of psoriasis [153].

A recent investigation discovered that the combination of recombinant mouse (rm) IL-18 with rmIL-23 exacerbates inflammation, upregulates IFN-γ and CXCL9, and enhances psoriasis-like epidermal hyperplasia [154]. In a mouse model, treatment with IL-18 in conjunction with rmIL-23 resulted in notable upregulation of CXCL9 expression compared to treatment with rmIL-23 alone. Furthermore, IL-17 showed an increase, although this change was not statistically significant. Conversely, IL-22 exhibited a non-significant decrease in the skin [154]. This suggests a potential cooperation between IL-18 and IL-23 in triggering a Th1 immune response, thereby worsening psoriatic inflammation. Additionally, IL-18 promotes the generation and persistence of Th17 cells [155]. Zhang et al. demonstrated that an IL-18-neutralizing antibody could impede the Th17 immune response in a psoriasis-like mouse model, indicating that the IL-18-mediated T cell response plays a crucial role in the pathogenesis of psoriasis and that inhibiting IL-18 could be a promising therapeutic strategy [49,155].

IL-18 levels are also elevated in the serum and stratum corneum of AD patients compared to healthy individuals [156,157,158,159]. These stratum corneum IL-18 levels strongly associate with barrier function alterations and AD severity [160]. In a transgenic mice study IL-18 was essential for developing AD-like dermatitis [161]. Chen et al. observed that IL-18 knockout mice exhibited lower inflammatory cell infiltration in AD-like lesions induced by MC903 compared to wild-type mice, although IgE levels were upregulated in the knockouts [162]. Additionally, IL-18 activates mast cells and basophils, inducing the expression of IL-4, IL-13, and histamine [162,163]. These data highlight the critical role of IL-18 in AD pathogenesis and its potential as a therapeutic target.

IL-37, in contrast, plays an important role reducing the inflammatory reaction in contact hypersensitivity. Luo et al. showed that increased IL-37 expression diminished auricular swelling and mitigated contact hypersensitivity in mice [138]. IL-37 expression in dendritic cells inhibits their antigen presentation and maturation ability while promoting regulatory T cells, thus suppressing contact hypersensitivity. In a murine model of AD induced by MC903, human IL-37b expression suppressed auricular swelling, pruritus, and the production of inflammatory cytokines and chemokines during AD progression [136,164]. Basophils, activated and recruited by inflammatory mediators like thymic stromal lymphopoietin (TSLP) produced by epithelial cells, release IL-4, promoting the differentiation of naïve T cells into Th2 cells and contributing to pruritus in AD [164,165,166]. Subcutaneous injections of human IL-37b reduced basophil infiltration and suppressed IL-4 production by basophils following TSLP stimulation, suggesting that IL-37 might improve AD through basophil modulation [164].

Eosinophils are involved in AD pathogenesis and contribute to the transition from a Th2-like immune response seen in acute AD lesions to a Th1-like immune response seen in chronic AD lesions [167]. IL-37 reduces eosinophilic inflammation [130,134,164].

Mast cells play a significant role in AD pathogenesis by suppressing production of IL-12 in dendritic cells, prompting a shift in T cell response towards a Th2 profile [167]. IL-33, a potent activator of mast cells, promotes their migration, maturation, adhesion, survival, and secretion of inflammatory cytokines [168]. IL-33-induced mast cell inflammation is mediated through the NF-kB and p38MAPK pathways [169]. IL-37, in conjunction with Smad3, inhibits NF-kB activation and p38MAPK phosphorylation, attenuating mast cell inflammation in AD [169].

Studies on IL-37 levels in serum and skin of AD patients have yielded controversial results. Fujita et al. reported elevated serum IL-37 levels that correlated with AD severity [121]. However, some patients with severe AD had reduced IL-37 levels, suggesting different AD phenotypes regarding IL-37 synthesis [121]. IL-37 levels in the skin were reduced in AD lesions compared to non-lesional skin [117,119]. Immunohistochemical analysis in healthy individuals identified IL-37 expression in the granular layer of the epidermis, along with loricrin [170] and filaggrin [117]. In AD patients, reduced IL-37 expression correlated with decreased filaggrin expression [117]. IL-33, a major cytokine in the pathogenesis of AD, may reduce IL-37 through two pathways: (a) downregulating loricrin and filaggrin expression either directly or through IL-33-induced IL-4 and IL-13 downregulation, and subsequently downregulating IL-37 [171,172,173], or (b) stimulating keratinocyte production of chemokines such as CXCL1 and CXCL8, promoting neutrophil infiltration, and CCL20, facilitating Th17 cell recruitment [174]. IL-17, whose release and production by neutrophils and Th17 cells is induced by IL-33 [122], would reduce filaggrin and loricrin expression and lead to IL-37 downregulation [171] (Figure 4).

IL-37 has also been investigated in psoriasis. In a mouse model, administering a plasmid containing human IL-37 improved skin lesions and decreased TNF expression [175]. Teng et al. reported that human IL-37 can inhibit the production of CXCL8, IL-6, and S100A7, all implicated in psoriasis pathogenesis. However, intradermal injection of IL-37 in a mouse model with imiquimod-induced psoriasis showed a protective effect without statistical significance, and lower levels of IL-37 in human psoriasis lesions compared to healthy skin [176]. Key cytokines involved in the pathogenesis of psoriasis, including IL-17A, IL-17C, IL-17F, and IL-22, may contribute to this phenomenon [176]. In randomized phase 2 trial, cutaneous levels of IL-37 in psoriasis lesions rapidly increased following treatment with tofacitinib [177].

### 6.1. IL-18 and IL-37 as Therapeutic Targets

Given their involvement in various inflammatory pathways, IL-18 and IL-37 have emerged as promising therapeutic targets. Multiple drugs targeting IL-18 are currently in development, though no specific drugs targeting IL-37 are under investigation at this time.

### 6.2. IL-18 as a Therapeutic Target

As IL-18 plays a pro-inflammatory role, drug development for inflammatory diseases typically focuses on inhibiting IL-18 [8]. Interestingly, inducers of IL-18 have also been studied as potential treatments for oncological conditions.

Currently, up to six different molecules targeting IL-18 are under development. Among these, three—Tadekinig alfa, CMK-389 and AMP18P1RA/IL-18bp-Fc-IL-1ra—are being specifically studied for dermatological conditions [56].

### 6.3. Tadekinig Alfa

Tadekinig alfa is a recombinant human IL-18 binding protein (IL-18BP) that inhibits IL-18. It is being studied for several conditions, including Adult-Onset Still’s Disease, lymphoproliferative disorders, immune system diseases, and macrophage activation syndrome [56]. In 2006, a phase 2 trial evaluated Tadekinig alfa in healthy volunteers and patients with psoriasis or rheumatoid arthritis. Unfortunately, it failed to demonstrate any efficacy in blocking IL-18 for these conditions [178].

### 6.4. CMK-389

CMK-389, an IL-18 inhibitor, was evaluated in a phase 2 randomized, placebo-controlled trial (NCT04836858) involving 71 patients with moderate to severe AD. Patients received either intravenous CMK389 10mg/kg (n = 34) or subcutaneous CMK389 300 mg (n = 17) or placebo (intravenous, n = 8, or subcutaneous, n = 8) every four weeks for a 12-week period. The endpoint was assessed at week 16 and an observational period followed, lasting approximately 12 weeks. The primary outcome, defined by the Investigator Global Assessment (IGA) response (clear or almost clear skin with at least a 2 point-reduction from baseline at week 16), was achieved by 14.7% of the intravenous CMK-389 group and 11.8% of the subcutaneous group, compared to none in the placebo groups. Serious adverse events, including AD worsening and heavy menstrual bleeding, were reported but did not lead to study discontinuation. Common adverse events included nasopharyngitis, COVID-19 infection, headache, and diarrhea.

### 6.5. AMP18P1RA/IL-18bp-Fc-IL-1ra

AMP18P1RA/IL-18bp-Fc-IL-1ra is a fusion protein consisting of IL-18BP and IL-1 receptor antagonist (IL-1ra) domains linked to the Fc fragment of human IgG1 [179]. The amino-terminal IL-18BP domain binds to and inhibits the activity of IL-18, whereas the carboxy-terminal IL-1ra binds to the IL-1 receptor (IL-1R) and antagonizes the effects of IL-1α and IL-1β, thus preventing their role in immune activation [180]. Currently, AMP18P1RA/IL-18bp-Fc-IL-1ra is in preclinical phases.

### 6.6. IL-37 as a Therapeutic Target

While no specific drugs targeting IL-37 are in development, the role of IL-37 in reducing inflammatory reactions in contact hypersensitivity and other conditions suggests potential therapeutic applications. In murine models, IL-37 has shown promise in suppressing inflammatory responses, suggesting the potential benefit of its modulation in treating inflammatory skin diseases.

## 7. Conclusions

This review highlights the complex involvement of IL-18 and IL-37 in the immune innate and adaptative systems and their roles in AD and psoriasis (Table 1). While significant advances have been made in understanding the pathogenesis of these diseases, particularly concerning IL-18, much remains unknown about IL-37. Despite the disappointing results of the first clinical trial evaluating Tadekinig alfa for psoriasis, a phase 2 trial of CMK-389 showed promising outcomes. Additionally, AMP18P1RA/IL-18bp-Fc-IL-1ra, with its unique mechanism of action, may also represent a promising future therapy.

## 8. Future Directions

Further research is needed to explore the therapeutic potential of targeting IL-18 and IL-37 in inflammatory skin diseases, with ongoing and future studies likely to provide deeper insights and new treatment avenues.

## Figures and Tables

**Figure 1 ijms-25-08437-f001:**
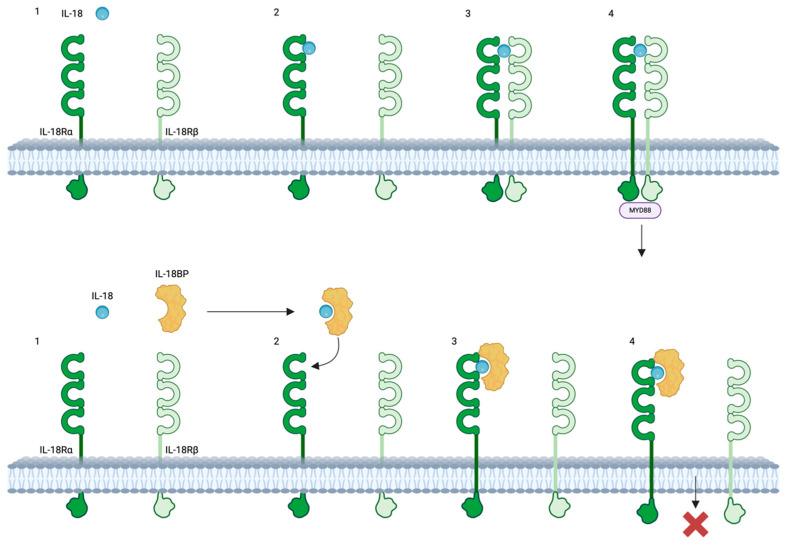
The interaction between IL-18, IL-18R and IL-18BP. Top: IL-18 initially binds to IL-18Rα, forming an inactive IL-18/IL-18Rα complex, which subsequently recruits the IL-18Rβ chain to establish a high-affinity complex that initiates the IL-18-dependent signaling pathway. Bottom: IL-18 binding protein (IL-18BP) binds to IL-18 and subsequently forms an inhibitory complex with IL-18Rα, thereby preventing IL-18Rβ from activating cells. Created with biorender.com. Abbreviations: IL-18, interleukin-18; IL-18BP, Interleukin-18 binding protein; IL-18Rα, Interleukin-18 receptor alpha; IL-18Rβ; interleukin-18 receptor beta.

**Figure 2 ijms-25-08437-f002:**
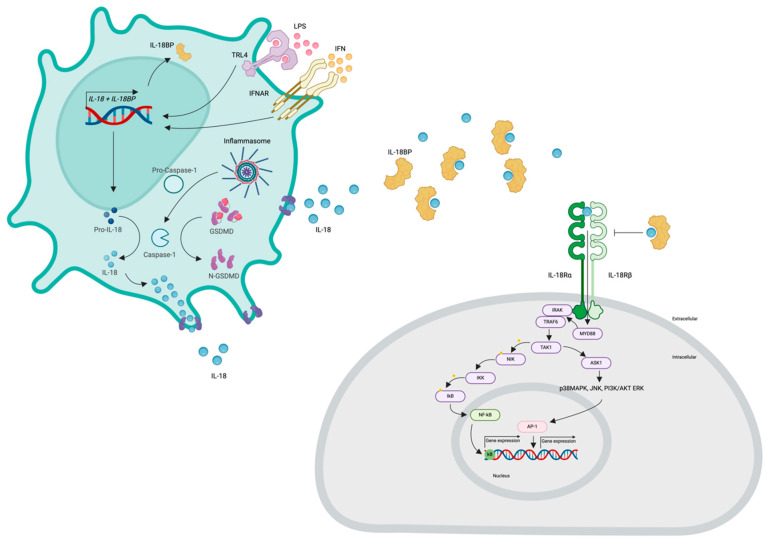
**IL-18 synthesis and signaling.** Top: Lipopolysaccharide and IFN induce the synthesis of pro-IL-18 and IL-18BP. Concurrently, the inflammasome activates pro-caspase-1 into caspase-1, which cleaves pro-IL-18 into IL-18. Additionally, caspase-1 cleaves gasdermin D (GSDMD) into N-GSDMD, forming pores through which IL-18 is released from the cells. Bottom: Once secreted into the extracellular space, IL-18 can bind to its receptor (IL-18R) or to IL-18 binding protein (IL-18BP). Binding to IL-18BP inhibits IL-18 signaling, whereas binding to IL-18R initiates the signaling cascade. The cytosolic domain of IL-18R contains a Toll/IL-1 receptor (TIR) domain that recruits Myd88, facilitating the recruitment of IL-1 receptor-associated kinase (IRAK). IRAK then associates with TNF receptor-associated factor 6 (TRAF6), which in turn binds to the kinase TAK1. TAK1 phosphorylates and activates NF-κB-inducing kinase (NIK). Activated NIK then activates the IκB kinase (IKK) complex, which phosphorylates IkB, leading to IkB ubiquitination and rapid degradation. This process liberates the NF-κB transcription factor, allowing it to translocate to the nucleus, where it binds to κB sites in the promoter regions of inflammatory genes. Moreover, TRAF6 phosphorylation also connects with apoptosis signal-regulating kinase 1 (ASK1) through TAK1, activating downstream mitogen-activated protein kinase (MAPK) signaling pathways, including p38 MAPK, c-Jun N-terminal kinase (JNK), phosphoinositide 3-kinase/protein kinase B (PI3K/AKT), and extracellulart signal-regulated kinase (ERK). This cascade culminates in the activation of the AP-1 transcription factor. Created with biorender.com. Abbreviations: LPS, lipopolysaccharide; IFN, interferon; GSDMD, gasdermin D; N-GSDMD, N-terminal gasdermin D, TIR, Toll/IL-1 receptor; Myd88, Myeloid differentiation primary response 88; IRAK, interleukin-1 receptor associated kinase; TRAF 6, TNF receptor-associated factor 6; TAK1, transforming growth factor-b-activated kinase 1; NIK, nuclear factor kappa B (NF-κB)-induced kinase; IKK, inhibitor of nuclear kappa B kinase; IkB, inhibitor of nuclear kappa B; NF-kB, nuclear factor kappa B; ASK1, apoptosis signal-regulating kinase 1; p38MAPK, p38 mitogen-activated protein kinase; JNK, Jun kinase; PI3K, phosphoinositide 3-kinase; AKT, protein kinase B; ERK extracellular regulated protein kinases; AP-1, activator protein 1.

**Figure 3 ijms-25-08437-f003:**
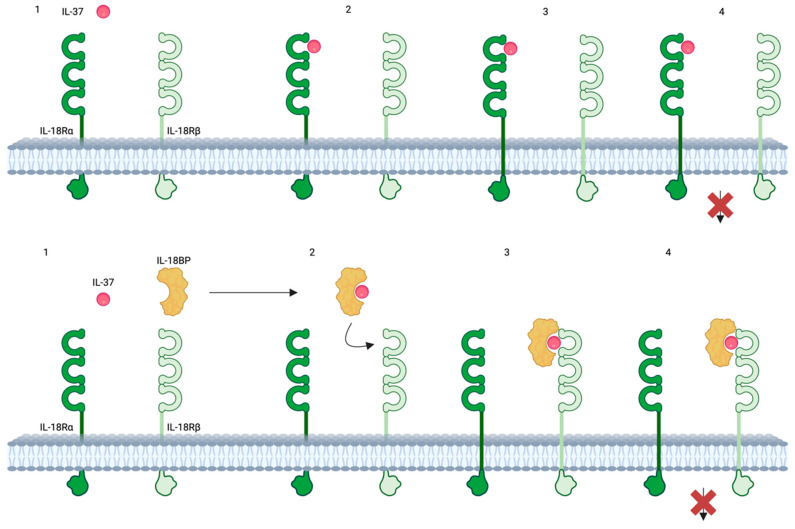
**IL-37 mechanism of action.** Top: IL-37 binds to IL-18Rα, causing the loss of recruitment with IL-18Rβ. Bottom: IL-37 binding to IL-18BP forms a complex with IL-18Rb, thereby enhancing the inhibition of IL-18 activity. Created with biorender.com. Abbreviations: IL-37, interleukin-37; IL-18BP, Interleukin-18 binding protein; IL-18Rα, Interleukin-18 receptor alpha; IL-18Rβ; Interleukin-18 receptor beta.

**Figure 4 ijms-25-08437-f004:**
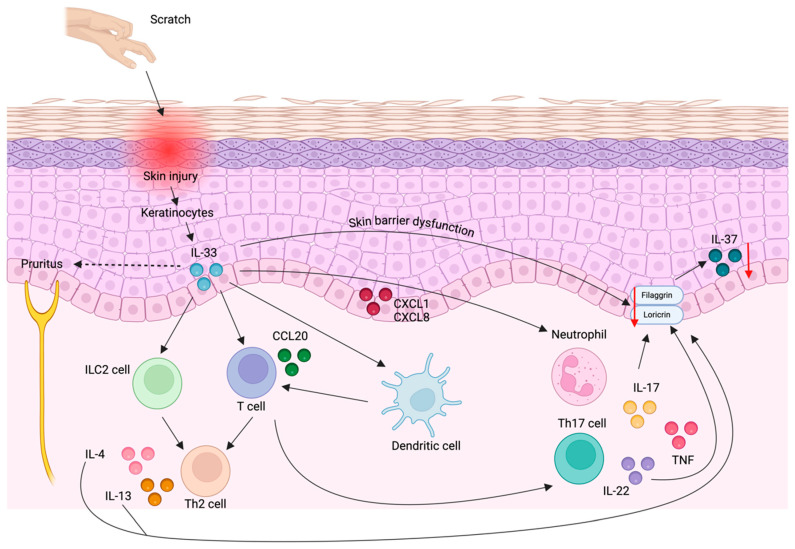
**IL-33 and IL-37 in the pathogenesis of psoriasis and AD.** Psoriasis and AD patients experiencing pruritus exhibit increased IL-33 production within keratinocytes following scratch injury. IL-33 subsequently activates pruritogenic sensory nerves, thereby promoting pruritus. Furthermore, IL-33 mediates a Th2 immune response in both psoriasis and AD through the activation of ILC2 cells, T cells, and dendritic cells. These Th2 cells produce IL-4 and IL-13, which downregulate filaggrin and loricrin, consequently reducing IL-37 expression in keratinocytes. Additionally, IL-33 induces the production of CCL20, which facilitates the recruitment of Th17 cells, and CXCL8, which promotes neutrophil influx into the epidermis. The accumulation of Th17 cells and neutrophils significantly contributes to psoriasis pathogenesis. Moreover, IL-33, along with IL-17 and IL-22, collectively downregulates filaggrin and loricrin expression, further reducing IL-37 expression in keratinocytes. Created with biorender.com. Abbreviations: IL-33, interleukin-33; ILC2 cells, innate lymphoid cells type 2; Th2 cell, T helper 2 cell; IL-4, interleukin-4; IL-13, interleukin-13; CCL20, C-C Motif Chemokine Ligand 20; Th17, T helper 17 cell; CXCL8, interleukin-8; IL-22, interleukin-22; TNF, tumor necrosis factor.

**Table 1 ijms-25-08437-t001:** Summary table of relevant studies related to IL-18 and IL-37.

Article	Important Findings
**IL-18**	
Bufler et al. (2002) [60]	IL-37b binds to IL-18BP, forming a complex with IL-18Rβ. This interaction prevents the β -chain from forming a functional receptor complex with IL-18Rα thereby inhibiting IL-18 activity
Landy et al. (2023) [76,89]	Signaling of the T cell receptor (TCR) is essential for IL-18 to interact with CD8+ T cells
Robinson et al. (1997) [76]	Interferon-gamma inducing factor (IGIF), unlike IL-12, does not initiate Th1 development but enhances IL-12-induced Th1 development in mice transgenic for an αβ TCR recognizing OVA There is a significant synergy between IGIF and IL-12 in promoting IFNγ production from developing and committed Th1 cells Both IL-12 and IGIF are necessary for the substantial expression of the Th1 phenotype
Xu et al. (2000) [90]	IL-18, either alone or in conjunction with IL-4, can stimulate T cells to differentiate into Th2 cells when T-cell receptor (TCR) activation occurs. This influence of IL-18 is predominantly exerted on CD4+ T cells rather than CD8+ T cells, and is suppressed in the presence of IL-12
Flisiak et al. (2006) [145]	There is an association between psoriasis severity and plasma IL-18 concentration
**IL-37**	
Hou et al. (2021) [164]	IL-37b reduces the expression of thymic stromal lymphopoietin (TSLP), the TSLP receptor, and basophil activation marker CD203c on basophils IL-4 release is reduced by IL-37 Treatment with a human IL-37b antibody in a murine model improves AD manifestations such as ear swelling and itching in a murine model
Guttman-Yassky et al. (2019) [120]	Significant downregulation of IL-37 correlates with skin barrier impairment in children with AD
Tsuji et al. (2022) [122]	IL-33 expression is increased by IL-37
Zhou et al. (2021) [117]	The addition of Th2 cytokines (IL-4, IL-13, and IL-31) to an EpidermFT in vitro 3D human skin model was shown to reduce epidermal IL-37 levels and promote the development of critical features of AD skin
Hou et al. (2021) [119]	Levels of IL-37 and its receptor IL-18R are significantly reduced in the blood of patients with AD There is a negative correlation between IL-37 and involucrin levels, and IL-37 inhibits involucrin expression in in vitro epidermal cell models.

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
