# Peer review of "A Narrative Review of the IL-18 and IL-37 Implications in the Pathogenesis of Atopic Dermatitis and Psoriasis: Prospective Treatment Targets"

_ijms, 2024, doi:10.3390/ijms25158437_

Round 1

Reviewer 1 Report

Comments and Suggestions for Authors

Authors present more a review on IL-18 and IL-37 than the role of those in dermatological disorders. The narrative side of the article requires significant changes and a lot of polishing. Despite very well written begining, article is mostly chaotic and in numerous parts it is hard to follow. Artile is accompanied by an overall well prepared and clear figures. Their description, however, should be mostly incorporated into main text. Most importantly, I am not sure where does the novelty of this review lay as there are several similar publications from recent years e.g. https://doi.org/10.3390/ijms24010372 https://doi.org/10.1016/j.biopha.2019.109705 https://doi.org/10.1371/journal.pone.0293327 https://doi.org/10.3389/fimmu.2023.955369 https://doi.org/10.3390/cells12232766

Some detailed comments below:

1. Lines 118-120: what is the nature of IL-18 - IL-37 direct interaction? The only interaction that I know of is the competion of IL-18 and IL-37 for IL-18RA. Please clarify

2. Please add the information about most important cellular source of IL-18.

3. Some parts are relatively hard to follow and should be re-written to increase the clarity. e.g. lines 222-234

4. Line 245: LCs are a subset of DCs. IL-18 stimulation affects the function of LCs, it may also affect other DCs, but is not that IL-18 stimulation of LCs causes the accumulation of other DCs.

5. The part 4.2. does not cover even the most basic information on IL-18 and immune cells, please see: http://dx.doi.org/10.18388/abp.2015_1153 and https://doi.org/10.3389/fimmu.2018.00763

6. IL-37 unlike IL-18 is secreted mostly (though not only) as a precursor, but this precursor is biologically active. Moreover, not all isoforms of IL-37 have the caspase 1 cleavage site. The general mechanism of production, secretion and action is well presented in 10.1111/imr.12605 Authors should re-write this section. Moreover, authors should discuss major cellular sources of IL-37.

7. Line 324 - how does Th2 response directly stimulate nerves?

8. Lines 343: what do authors mean by "through PBMCs"?

9. The modes of IL-37 action on immune cells is chaotic and should be significantly changed and reorganized. It is also unclear why authors provide numerous information about IL-37 in atopic dermatitis in this paragraph while this belongs to the next (6.).

10. Line 396 - the fact that IL-18 accumulates in psoriatic skin does not warrant that it may be considered as a biomarker. How specific is it? Does it happen in other skin disorders as well? Does it happen in each and every psoriatic case? "A potential biomarker" is one widely over-utilized expression, there are numerous "potential biomarkers" being suggested each month, but almost none of those enter clinical practice. Most commonly because of very low specificity.

11. Lines 402-209: IL-23 is an important cytokine to promote Th17 response. IL-17 is one of the disregulated cytokines in psoriasis. It is not unexpected that recombinant IL-23 would exacerbate psoriasis in mouse model.

Comments on the Quality of English Language

Technically language is mostly acceptable, but some significant changes are required to increase the clarity and flow of the narration

Author Response

Reviewer 1:

Authors present more a review on IL-18 and IL-37 than the role of those in dermatological disorders. The narrative side of the article requires significant changes and a lot of polishing. Despite very well written beginning, article is mostly chaotic and in numerous parts it is hard to follow. Article is accompanied by an overall well prepared and clear figures. Their description, however, should be mostly incorporated into main text. Most importantly, I am not sure where does the novelty of this review lay as there are several similar publications from recent years e.g. https://doi.org/10.3390/ijms24010372 https://doi.org/10.1016/j.biopha.2019.109705 https://doi.org/10.1371/journal.pone.0293327 https://doi.org/10.3389/fimmu.2023.955369 https://doi.org/10.3390/cells12232766

Thank you for your thorough review; we greatly appreciate your comments. We have slightly modified the structure of the manuscript to improve clarity and comprehension. As a narrative review, we have examined an extensive list of manuscripts, including those mentioned. However, none of the provided manuscripts offers a comprehensive review of both interleukins and their roles in psoriasis and atopic dermatitis.

Some detailed comments below:

1. Lines 118-120: what is the nature of IL-18 - IL-37 direct interaction? The only interaction that I know of is the competition of IL-18 and IL-37 for IL-18RA. Please clarify

Thank you for your comment. The reviewer is right above the absence of a direct interaction between IL-37 and IL-18 other than competition for IL-18R. We have modified the text: “IL-37 also influences extracellular IL-18 function, dampening its inflammatory effects, by binding to IL-18R

  1. Please add the information about most important cellular source of IL-18.

We have added the following sentence (line 101-102): “IL-18 is a cytokine derived from macrophages, and nearly all barrier epithelia harbor a significant reservoir of pro-IL-18 [56].”

  1. Some parts are relatively hard to follow and should be re-written to increase the clarity. e.g. lines 222-234

Thank you for your suggestions. We have revised the paragraph to enhance clarity: “IFN-γ induces the expression of chemokines such as CXCL9 (chemokine C-X-C motif ligand 9), CXCL10, and CXCL11. These chemokines recruit Th1 cells by binding to CXCR3 on the cell surface, resulting in significant infiltration of Th1 cells observed in inflammatory skin conditions [96]. In keratinocytes, IL-18 amplifies the mRNA expression of CXCL9, CXCL10, and CXCL11 induced by IFN-γ [97]. Additionally, IFN-γ activates signal transducer and activator of transcription 1 (STAT1) via Janus kinase 1 (JAK1)/JAK2 and/or p38MAPK pathways, leading to the production of CXCL9, CXCL10, and CXCL11. IFN-γ may also influence the activation of interferon regulatory factor-1 (IRF-1), responsible for CXCL11 expression, primarily through the p38MAPK pathway. IL-18 enhances this activation via PI3K/AKT and MEK/ERK pathways. IL-18 induces NF-kB activity through MEK/ERK and PI3K/AKT pathways, thereby augmenting IFN-γ -induced CXCL9 secretion.

  1. Line 245: LCs are a subset of DCs. IL-18 stimulation affects the function of LCs, it may also affect other DCs, but is not that IL-18 stimulation of LCs causes the accumulation of other DCs.

Thank you for your suggestions. This sentence has been amended: “Upon stimulation by diverse triggers, LCs undergo activation into mature DCs. Specifically, IL-18 induces the accumulation of DCs and the migration of LCs to local lymph nodes”.

5. The part 4.2. does not cover even the most basic information on IL-18 and immune cells, please see:
http://dx.doi.org/10.18388/abp.2015_1153 and https://doi.org/10.3389/fimmu.2018.00763

Thank you for your suggestions. We have added the following information: “Also, IL-18 has been demonstrated to contribute to the induction of Th17 cell responses. It has been proposed that IL-18 enhances IL-17 production in already polarized Th17 cells in conjunction with IL-23, independently of TCR activation.

Kinoshita et al. observed that IL-18 injections activated B-1 cells in the liver, leading to increased production of IgM and enhanced immunity against bacterial infections following a burn injury [107].

Finally, the role of IL-18 in maintaining homeostasis is underscored by findings from studies involving IL-18 deficient mice, which demonstrated a predisposition to obesity and other metabolic disorders. These mice exhibited a substantial increase in body weight (by 40%) and body fat content (over 100%) compared to wildtype animals. Similarly, individuals with impaired expression of the IL-18Rα receptor on their cell surfaces also displayed susceptibility to obesity, diabetes, and other metabolic disorders [106,108]

Additional information concerning the role of IL-18 in immune cells, previously discussed, has been relocated to section 4.2.

6. IL-37 unlike IL-18 is secreted mostly (though not only) as a precursor, but this precursor is biologically active. Moreover, not all isoforms of IL-37 have the caspase 1 cleavage site. The general mechanism of production, secretion and action is well presented in 10.1111/imr.12605 Authors should re-write this section. Moreover, authors should discuss major cellular sources of IL-37.

Thank you for your suggestions. We have modified our section on IL-37: “IL-37 expression has been documented in multiple human tissues and cell lines. IL-37 is mainly produced by circulating monocytes, macrophages, dendritic cells, tonsillar B cells, and plasma cells, as well as by epithelial cells in the skin and gut as a response to inflammation.

Despite structural and production process similarities between IL-37 and other cytokines of the IL-1 family, particularly IL-18, notable differences exist. IL-37 is secreted as a precursor form that is biologically active both in vitro and in vivo. This active precursor form of IL-37 is mainly secreted into the extracellular space. Conversely, the intracellular precursor form of IL-37 is activated by caspase-1-mediated cleavage, allowing its translocation into the nucleus.”

  1. Line 324 - how does Th2 response directly stimulate nerves?

Here is a more detailed explanation: “IL-33 plays a well-established role in the pathogenesis of AD by inducing a type 2 immune response, which produces IL-31 that directly stimulates nerves, leading to pruritus [124]. According to Cevikbas et al. [125], IL-31 production in skin cells was found to be primarily attributed to Th2 cells, with a lesser contribution from mature dendritic cells among the immune and resident skin cell populations studied. Injection of IL-31 into the skin and spinal cord induced severe itching, and its levels notably increased in murine models of atopy-like dermatitis. Both human and mouse dorsal root ganglia neurons expressed IL-31RA, particularly in neurons co-expressing transient receptor potential cation channel vanilloid subtype 1 (TRPV1). Itch induced by IL-31 was significantly di-minished in mice deficient in TRPV1 and transient receptor potential cation channel ankyrin subtype 1 (TRPA1), but not in c-kit or proteinase-activated receptor 2 mice. In vitro studies with cultured primary sensory neurons revealed that IL-31 triggered calcium release and phosphorylation of extracellular signal-regulated kinase 1/2 (ERK1/2), inhi-bition of which disrupted IL-31 signaling and reduced IL-31-induced scratching in vivo” (Cevikbas et al. DOI: 10.1016/j.jaci.2013.10.048).

  1. Lines 343: what do authors mean by "through PBMCs"?

Through PBMCs means - by, using or via – peripheral blood mononuclear cells (PBMCs).

  1. The modes of IL-37 action on immune cells is chaotic and should be significantly changed and reorganized. It is also unclear why authors provide numerous information about IL-37 in atopic dermatitis in this paragraph while this belongs to the next (6.).

Thank you for your suggestions, we have reorganized paragraphs of IL-37 action in immune cells and part 6. Information about IL-37 in AD has been re-located to paragraph 6.

  1. Line 396 - the fact that IL-18 accumulates in psoriatic skin does not warrant that it may be considered as a biomarker. How specific is it? Does it happen in other skin disorders as well? Does it happen in each and every psoriatic case? "A potential biomarker" is one widely over-utilized expression, there are numerous "potential biomarkers" being suggested each month, but almost none of those enter clinical practice. Most commonly because of very low specificity.

Thank you for your appreciation. As you imply, IL-18 is not specific of psoriasis, and it is still not known if it occurs in every case. IL-18 has also been documented as a potential biomarker of rheumatoid arthritis, systemic lupus erythematosus, inflammatory bowel disease, insulin resistance and metabolic syndrome, cardiovascular conditions, and various cancers (DOI: 10.1172/JCI7317; DOI: 10.1191/096120300678828703; DOI: 10.2337/diacare.26.5.1647; DOI: 10.1161/hh1901.098735; DOI: 10.1189/jlb.5RU0714-360RR).

Similarly, a biomarker can be used to diagnose a disease, but it can also be utilized to monitor disease progression and even detect subclinical inflammation. Therefore, IL-18 could serve as a biomarker of inflammation, as perfectly emphasized in the reference you suggested: (https://doi.org/10.1371/journal.pone.0293327).

It is the responsibility of clinicians to attribute this inflammation to a specific diagnosis, which could be psoriasis or atopic dermatitis.

We have added the following sentence: “Nonetheless, IL-18 levels have also been suggested as potential biomarkers for multiple diseases, including: rheumatoid arthritis [151], systemic lupus erythematosus [152], inflammatory bowel disease [153], insulin resistance and metabolic syndrome [154], atherosclerosis, myocardial infarction and other cardiovascular conditions [155], as well as various cancers such as colorectal cancer, and non-small cell lung cancer [156]. Therefore, although IL-18 is not specific to psoriasis nor the sole biomarker for the condition, its increased levels in serum of patients with psoriasis suggests that IL-18 may be useful in diagnosing, monitoring, and understanding the pathogenesis of psoriasis.”

  1. Lines 402-209: IL-23 is an important cytokine to promote Th17 response. IL-17 is one of the dysregulated cytokines in psoriasis. It is not unexpected that recombinant IL-23 would exacerbate psoriasis in mouse model.

According to Shimoura et al. (DOI: 10.1007/s00403-017-1735-2), the mouse model treated with IL-18 and rmIL-23 showed significant upregulation of CXCL9 expression compared to rmIL-23 alone. Additionally, IL-17 exhibited an increase, although it was not statistically significant. In contrast, IL-22 demonstrated a non-significant decrease in its levels.

We have added the following sentence: “A recent investigation discovered that the combination of recombinant mouse (rm) IL-18 with rmIL-23 exacerbates inflammation, upregulates IFN-γ and CXCL9, and enhances psoriasis-like epidermal hyperplasia [158]. In a mouse model, treatment with IL-18 in conjunction with rmIL-23 resulted in notable upregulation of CXCL9 expression compared to treatment with rmIL-23 alone. Furthermore, IL-17 showed an increase, although this change was not statistically significant. Conversely, IL-22 exhibited a non-significant decrease in the skin [158]

Reviewer 2 Report

Comments and Suggestions for Authors

The document provides a comprehensive review of the roles of IL-18 and IL-37 in the pathogenesis of atopic dermatitis (AD) and psoriasis, highlighting their potential as therapeutic targets. The review synthesizes current knowledge on these interleukins, their involvement in inflammatory processes, and emerging treatment strategies.

Suggestions:

1. The abstract could be more concise. It currently includes detailed results that may be better suited for the main body of the text.

2. Break down the "Biology of IL-18" and "Biology of IL-37" into subsections that detail their synthesis, signaling, and roles in diseases.

3. Including a brief comparison with other cytokines involved in AD and psoriasis could provide context and highlight the unique roles of IL-18 and IL-37

4. Some references are not properly formatted or are incomplete

5. Terms should be used consistently throughout the document. For instance, IL-18 and IL37 should always be formatted the same way (either IL-18 and IL-37 or IL18 and IL37).

6. Some sections contain redundant information. For example, the introduction and sections on IL-18 and IL-37 might have overlapping content about their roles in AD and psoriasis.

7. Introduction is too short.  Provide a broader context about the prevalence and impact of atopic dermatitis and psoriasis globally; highlight their chronic nature, the burden on patients, and the limitations of current treatment options.

8. There are several instances of misplaced or duplicated numbers and symbols throughout the text

Comments on the Quality of English Language

There are several instances of misplaced or duplicated numbers and symbols throughout the text

Author Response

The document provides a comprehensive review of the roles of IL-18 and IL-37 in the pathogenesis of atopic dermatitis (AD) and psoriasis, highlighting their potential as therapeutic targets. The review synthesizes current knowledge on these interleukins, their involvement in inflammatory processes, and emerging treatment strategies.

Suggestions:

  1. The abstract could be more concise. It currently includes detailed results that may be better suited for the main body of the text.

Thank you for your suggestions, we have amended the abstract.

  1. Break down the "Biology of IL-18" and "Biology of IL-37" into subsections that detail their synthesis, signaling, and roles in diseases.

Thank you for your comments; we have created the subsections.

  1. Including a brief comparison with other cytokines involved in AD and psoriasis could provide context and highlight the unique roles of IL-18 and IL-37

Thank you for your response. We agree that a deeper exploration of the cytokine networks involved in the pathogenesis of psoriasis and atopic dermatitis would offer valuable comparison and context to the roles of IL-18 and IL-37, though it might be too extensive for the current scope of our manuscript.

  1. Some references are not properly formatted or are incomplete

We have amended the references.

  1. Terms should be used consistently throughout the document. For instance, IL-18 and IL37 should always be formatted the same way (either IL-18 and IL-37 or IL18 and IL37).

Thank you for your response. Errors have been identified and corrected. It should be noted that the correct designation for the IL-18 gene is IL18, without hyphen.

  1. Some sections contain redundant information. For example, the introduction and sections on IL-18 and IL-37 might have overlapping content about their roles in AD and psoriasis.

Thank you for your commentaries. We have attempted to correct this issue following your recommendations.

  1. Introduction is too short.  Provide a broader context about the prevalence and impact of atopic dermatitis and psoriasis globally; highlight their chronic nature, the burden on patients, and the limitations of current treatment options.

Thank you for your response. We have added the following sentence: “Psoriasis and atopic dermatitis (AD) are chronic immune-mediated inflammatory skin conditions that collectively affect a substantial proportion of the population and significantly diminish quality of life, particularly impacting mental well-being [11,12]. These conditions despite advancements in the management of psoriasis and AD, considerable gaps persist in identifying optimal treatment strategies tailored to individual patient needs.”

  1. There are several instances of misplaced or duplicated numbers and symbols throughout the text

Thank you for your response, we have amended the errors detected.

Reviewer 3 Report

Comments and Suggestions for Authors

Dear Authors, 

This paper is a review o the latest publications regarding atopic dermatitis and psoriasis are prevalent inflammatory skin conditions with diverse treatment options.

This paper discuss the topic of Atopic Dermatitis and Psoriasis Pathogenesis, Biology of IL-18, IL-18 in Keratinocytes, IL-18 in immune Cells, Biology of IL-37, iL-37 in the Skin, IL-37 in Immune Cells and IL-18 and IL-37 in Inflammatory Skin Diseases.

I think this paper is missed a summary table, statistical data and Prisma analysis.

 Thank you 

Sincerely.

Author Response

This paper is a review of the latest publications regarding atopic dermatitis and psoriasis are prevalent inflammatory skin conditions with diverse treatment options.

This paper discusses the topic of Atopic Dermatitis and Psoriasis Pathogenesis, Biology of IL-18, IL-18 in Keratinocytes, IL-18 in immune Cells, Biology of IL-37, IL-37 in the Skin, IL-37 in Immune Cells and IL-18 and IL-37 in Inflammatory Skin Diseases.

I think this paper is missed a summary table, statistical data and Prisma analysis.

Thank you for your suggestions; however, this paper is a narrative review, so neither statistical data nor a PRISMA analysis are needed.

Table 1: Summary table of relevant studies related to IL-18 and IL-37

Article

Important findings

IL-37

Hou et al. (2020)

-        IL-37b significantly increased Foxp3+ regulatory T cells (Treg) and IL-10 levels, while also decreasing eosinophil infiltration in ear lesions

Hou et al. (2021)

-        IL-37b reduced the expression of thymic stromal lymphopoietin (TSLP), the TSLP receptor, and basophil activation marker CD203c on basophils

-        IL-4 release was reduced by IL-37

-        Treatment with a human IL-37b antibody, AD symptoms such as ear swelling and itching improved

-         

Guttman-Yassky et al. (2019)

-        Skin barrier of children with AD showed significant downregulation of IL-37

Tsuji et al. (2022)

-        IL-33 expression was increased by IL-37

Zhou et al. (2021)

-        Th2 cytokines added to the skin model was sufficient to reduce epidermal IL-37 levels and acquire critical features of AD skin

Hou et al. (2021)

-        AD patients exhibited significantly reduced levels of IL-37 and its receptor IL18R

-        There was a negative correlation found between IL-37 and involucrin, and IL-37 was demonstrated to inhibit involucrin expression in in vitro epidermal cell models.

IL-18

Bufler et al. (2002)

-        IL-37b binds to IL-18BP, forming a complex with IL-18Rβ. This interaction prevents the β -chain from forming a functional receptor complex with IL-18Rα thereby inhibiting IL-18 activity

Landy et al. (2023)

-        Signaling of the T cell receptor (TCR) is essential for IL-18 to interact with CD8+ T cells

Robinson et al. (1997)

-        IGIF, in contrast to IL-12, does not initiate Th1 development but enhances IL-12-induced Th1 development

-        There is a significant synergy between IGIF and IL-12 in promoting IFNγ production from developing and committed Th1 cells

-        Both IL-12 and IGIF are necessary for the substantial expression of the Th1 phenotype

Xu et al. (2000)

-        L-18 can stimulate T cells to differentiate into Th2 cells when TCR activation occurs, either alone or in conjunction with IL-4.

-        This influence of IL-18 is predominantly exerted on CD4+ T cells rather than CD8+ T cells, and it is suppressed in the presence of IL-12

Flisiak et al. (2006)

-        There is an association between psoriasis severity and plasma IL-18 concentration

Forouzandeh et al. (2020)

-        Patients with psoriasis exhibit higher serum levels of IL-18, compared to controls

Reviewer 4 Report

Comments and Suggestions for Authors

The article may be an useful contribution to the journal; however, few changes should be taken into consideration, in the benefit of the reader:

The Methodology section does not contain a PRIMSA graph and criteria for selecting articles in database search that the authors have been provided. Therefore, it must be specified if the review was systematic (inclusion/exclusion criteria, strict methodology) or just a narrative review. In the last case, of a narrative review, it must be specified clearly/written in a clear manner in Methodology section and the title should include “…-a narrative review” to respect scientific writing standards, in the interest if the reader.

The reason for presenting the role of the selected interleukins in specifically AD and psoriasis, side by side, is not evident from the Abstract; it should be clarified why the researchers chose these two skin conditions, while they did not include other skin conditions. Moreover, the abstract should be rewritten in the interest of the reader, to make clear how these ILs establish a connection between the 2 skin diaseases that are so different in terms of pathogenesis. Also, this connection should be mirrored in the body of the manuscript.

Grammar and punctuation must also be carefully checked within the entire article .

Comments on the Quality of English Language

minor

Author Response

The article may be a useful contribution to the journal; however, a few changes should be taken into consideration, in the benefit of the reader:

The Methodology section does not contain a PRIMSA graph and criteria for selecting articles in database search that the authors have been provided. Therefore, it must be specified if the review was systematic (inclusion/exclusion criteria, strict methodology) or just a narrative review. In the last case, of a narrative review, it must be specified clearly/written in a clear manner in Methodology section and the title should include “…-a narrative review” to respect scientific writing standards, in the interest if the reader.

Thank you for your response. This manuscript is a narrative review. We have amended the title: “Narrative review of the IL-18 and IL-37 Implications in the Pathogenesis of Atopic Dermatitis and Psoriasis: Prospective Treatment Targets

The reason for presenting the role of the selected interleukins in specifically AD and psoriasis, side by side, is not evident from the Abstract; it should be clarified why the researchers chose these two skin conditions, while they did not include other skin conditions. Moreover, the abstract should be rewritten in the interest of the reader, to make clear how these ILs establish a connection between the 2 skin diseases that are so different in terms of pathogenesis. Also, this connection should be mirrored in the body of the manuscript.

Thank you for your suggestions, we have added information in the abstract: “Atopic dermatitis and psoriasis are prevalent inflammatory skin conditions that significantly impact the quality of life of patients, with diverse treatment options available. Despite advances in understanding their underlying mechanisms, recent research highlights the significance of interleukins IL-18 and IL-37, in Th1, Th2, and Th17 inflammatory responses, closely associated with the pathogenesis of psoriasis and atopic dermatitis. Hence, IL-18 and IL-37 could potentially become therapeutic targets.”

Grammar and punctuation must also be carefully checked within the entire article.

Thank you, we have revised the grammar and punctuation throughout our manuscript.

Round 2

Reviewer 1 Report

Comments and Suggestions for Authors

I am generally satisfied with author responses, my major comments were sufficiently addressed. 

Comments on the Quality of English Language

Some changes required. Language is understandable, but could be enhanced. 

Author Response

I am generally satisfied with author responses, my major comments were sufficiently addressed. 

Thank you for your revision.

Some changes required. Language is understandable, but could be enhanced

We have modified the introduction section: ". Psoriasis and atopic dermatitis (AD) are chronic immune-mediated inflammatory skin conditions affecting a significant proportion of the world’s population. They carry a significant disease burden, diminishing the health-related quality of life and particularly impacting mental well-being [11,12]. Despite significant advancements in the targeted treatment of psoriasis and AD through the inhibition of key cytokines with pathogenetic relevance, considerable gaps persist in identifying optimal treatment strategies tailored to individual patients’ needs. Growing insights into the roles of IL-18 and IL-37 in these diseases may contribute to the development of newer therapeutic alternatives. "

Reviewer 4 Report

Comments and Suggestions for Authors

The manuscript bas been improved. 

Comments on the Quality of English Language

Minor editing required. 

Author Response

The manuscript bas been improved. 

Thank you for your revision.

Minor editing required. 

We have modified the introduction section: ". Psoriasis and atopic dermatitis (AD) are chronic immune-mediated inflammatory skin conditions affecting a significant proportion of the world’s population. They carry a significant disease burden, diminishing the health-related quality of life and particularly impacting mental well-being [11,12]. Despite significant advancements in the targeted treatment of psoriasis and AD through the inhibition of key cytokines with pathogenetic relevance, considerable gaps persist in identifying optimal treatment strategies tailored to individual patients’ needs. Growing insights into the roles of IL-18 and IL-37 in these diseases may contribute to the development of newer therapeutic alternatives. "